# ProAKAP4 Concentration Is Related to Sperm Motility and Motile Sperm Subpopulations in Frozen–Thawed Horse Semen

**DOI:** 10.3390/ani12233417

**Published:** 2022-12-04

**Authors:** Marta Dordas-Perpinyà, Ivan Yanez-Ortiz, Nicolas Sergeant, Vincent Mevel, Jean-François Bruyas, Jaime Catalán, Maryse Delehedde, Lamia Briand-Amirat, Jordi Miró

**Affiliations:** 1Oniris, Nantes-Atlantic College of Veterinary Medicine, 44300 Nantes, France; 2Equine Reproduction Service, Department of Animal Medicine and Surgery, Faculty of Veterinary Sciences, Autonomous University of Barcelona, 08193 Cerdanyola del Vallès, Spain; 3CHRU of Lille, University of Lille, INSERM, UMRS-1172, 59045 Lille, France; 4SPQI S.A.S, 59000 Lille, France

**Keywords:** ProAKAP4, AKAP4, stallion, spermatozoa, spermatic subpopulations

## Abstract

**Simple Summary:**

ProAKAP4 is the precursor of AKAP4 protein, whose principal function is to anchor protein complexes to regulate the long-term motion and flagellum structure of mammal spermatozoa. The objective of the present study was to evaluate how proAKAP4 sperm concentrations correlate with the motion parameters of equine spermatozoa. We showed that proAKAP4 concentrations are associated with motility and velocity parameters. Low proAKAP4 concentrations reflect a loss of spermatozoa functionality. In conclusion, proAKAP4 concentrations are associated with better motion parameters in stallions.

**Abstract:**

ProAKAP4 is the precursor of AKAP4 (A-kinase Anchor protein 4), the main structural protein of the fibrous sheath of sperm. The amount of proAKAP4 reflects the ability of spermatozoa to maintain the flagellum activity and functionality up to the site of fertilization and is positively correlated with progressive motility in several mammalian species. The aim of this study was to investigate the relationship between proAKAP4 concentration with horse sperm motility descriptors and spermatic motile subpopulations. For this purpose, a total of 48 ejaculates from 13 different stallions were analyzed. Spermatic motility descriptors were obtained by the CASA system, and four motile subpopulations (SP) with specific motility patterns were statistically identified. ProAKAP4 concentrations were evaluated by ELISA. The relationship between motility descriptors of sperm subpopulations and proAKAP4 concentrations was evaluated. Following a hierarchical cluster statistical analysis, ejaculates were divided into two groups according to their proAKAP4 concentrations, either having low proAKAP4 concentrations (5.06–35.61 ng/10M spz; n = 23) or high (39.92–82.23 ng/10M spz; n = 25) proAKAP4 concentrations (*p* < 0.001). ProAKAP4 concentrations were positively correlated (*p* < 0.05) with total and progressive motility, as well as with parameters of velocity. ProAKAP4 amount also showed a negative correlation (*p* < 0.05) with sperm motile subpopulation number 3, which was the subpopulation with the lowest velocity parameters. In conclusion, proAKAP4 concentration in stallion semen positively reflects sperm progressive motility with the functional velocity kinematic descriptors. Concentrations of proAKAP4 higher than 37.77 ng/10M spz were correlated with a very good quality frozen/thawed stallion semen.

## 1. Introduction

Spermatozoa are basic motile cells to provide fertilization capacity. Formerly, sperm analytics was subjective, but since the late 1980s, when the first CASA (computer-assisted sperm analysis) was commercialized [1], motion kinematic descriptors and morphometric characteristics have been analyzed objectively [2]. The use of the CASA system evinced sperm motile subpopulation structures inside the stallion ejaculate with specific motility patterns [3,4]. These subpopulations originate during spermatogenesis. They show different phenotypic characteristics in motility parameters [5] when the quality of each cell is affected. The simplest interpretation of the subpopulation structure is that groups represented different levels of sperm quality values given by CASA [3,4]. Spermatic subpopulations are maintained despite the treatment (centrifugation or frozen/thawed), and the sole parameter which modifies them is the proportion between them [6,7]. Different subpopulations, 3 or 4, have been identified in different mammals, birds, or salmonids [8]. It seems that the equilibrium in the subpopulation structure could be related to fertility. In the study by Quintero-Moreno [3], the percentage of spermatozoa included in the subpopulation with excellent motility patterns was related to fertility.

Spermatozoa motility is driven by the flagellum which is a highly organized organelle. The whole flagellum has an axoneme along the centre, surrounded in the principal piece by a fibrous sheath. The principal piece drives spermatozoa motility. Whereas the proximal part of the flagellum, the so-called intermediate piece, contains mitochondria around it, the distal or terminal piece mainly consists of the axoneme. A crosstalk between the spermatozoa environment, the fibrous sheath, and the axoneme are necessary for the long-lasting maintenance of sperm motility and capacitation. The fine tuning mechanism necessitates a coordinated and localized signalling machinery to deliver the proper signal and regulatory process to regulate sperm motility. ProAKAP4 is the precursor of AKAP4 (A-kinase Anchor protein 4), which is an essential and more abundant structural protein of the fibrous sheath of the principal piece of the flagellum [9,10,11,12]. ProAKAP4 and its active form, AKAP4, are expressed in a variety of mammalian species, and they are highly conserved (~70%) in the animal kingdom [9]. They are present in the horse [13], bull [14], pig [15], ram [16], koala [17], crocodile [18], camel [19] and dog [20,21,22]. ProAKAP4 and AKAP4 constitute a unique fibrillar structure never observed in other flagellated cell types [23]. They are intracellular proteins incorporated in the columns and ribbons of the fibrous sheath of the spermatozoa flagellum [11] and are never found in seminal plasma [19,24]. ProAKAP4 appears at the spermatic stage during spermatogenesis, becoming a storage form of the AKAP4 [25]. AKAP4 exerts an impact on the specificity of the transduction signal and metabolic processes that support sperm motility, hypermotility, and capacitation [9,10,11]. ProAKAP4 amount reflects the ability of spermatozoa to keep the flagellum active and functional, up to the site of fertilization [26,27], being positively correlated with progressive motility in several mammalian species [9,10,28]. Targeted disruption of the AKAP4 gene produces knock-out animals with spermatozoa with sperm fibrous sheath dysplasia and failure of progressive motility [26,27].

In horses, proAKAP4 was described for the first time by Turner [13] as an inactive form of AKAP4. The latter is the protein form that anchors A-protein kinase and regulates the AMPc-dependent signalling specifically localized to spermatozoa flagellum. This original [13] suggests that the AKAP4 function relates to the motility control mechanisms of the spermatozoa. Recently, Blommaert et al. [24] showed a correlation between proAKAP4 concentrations and total and progressive motility. Interestingly proAKAP4 was significantly more abundant in high-quality spermatozoa isolated using density gradient centrifugation in cross-bred stallions, and AKAP4 expression was significantly correlated with motility parameters and fertility [28]. Furthermore, the initial proAKAP4 amount was shown to be different when post-thawed stallions’ semen was stored in two types of extenders [29]. ProAKAP4 could be a good biomarker of semen quality in stallions due to its correlation with progressive motility, as well as its expression in more mature spermatozoa [28].

The aim of this study is to evaluate the relationship between proAKAP4 concentration and motility descriptors, as well as motility descriptor-based subpopulations of spermatozoa in stallions in frozen/thawed horse semen.

## 2. Materials and Methods

### 2.1. Semen Collection and Cryopreservation

The collection of the samples was performed at the Equine Reproduction Service of the Universitat Autonoma de Barcelona (Bellaterra, Cerdanyola del Vallès, Spain), which is a European centre approved to produce semen with authorization code ES09RS01E. Stallions used came at the service for commercial semen freezing. Thirteen stallions from 4 to 18 years old of several breeds (Arabian, Andalusian, and Warmblood) were used, with a total of 48 ejaculates kept counting between 1 to 5 ejaculates per stallion.

Semen collection was conducted using a Hannover artificial vagina (Minitüb GmbH, Tiefen-bach, Germany). The ejaculate without gel was diluted 1:5 in Kenney extender in 50 mL corning tubes previously warmed at 37 °C. All ejaculates were evaluated before freezing: total volume was recorded, sperm concentration was evaluated by a haemocytometer (Neubauer chamber, Paul Marienfeld GmbH and Co., KG; Lauda-Königshofen, Germany), motility by a CASA system (Section 2.2), and sperm morphology and viability using eosin-nigrosin staining [30].

### 2.2. Sperm Cryopreservation

Prior to cryopreservation, each extended semen sample was centrifuged at 660× *g* and 20 °C for 15 min (Medifriger BL-S, JP Selecta S.A., Barcelona, Spain) to remove seminal plasma. Supernatants were discarded, and pellets were resuspended in a commercial freezing medium (Botucrio^®^, Botupharma Animal Biotechnology; Botucatu, Brazil). Afterwards, sperm concentration and viability were re-evaluated, and freezing medium (Botucrio **^®^**) was added to obtain a final concentration of 200 × 106 per mL of viable sperm. Samples were packaged into 0.5 mL straws and cryopreserved using a controlled-rate freezer (Ice-Cube 14S; Minitüb). Briefly, the cryopreservation procedure consists of three successive freezing steps: first: cooling from 20 °C to 5 °C at rate of −0.25 °C/min for 60 min; second: freezing from 5 °C to −90 °C at a rate of −4.75 °C/min for 20 min; and third: freezing step from −90 °C to −120 °C at a rate −11.11 °C/min for 2.7 min. Straws were stored into liquid nitrogen in appropriate tanks.

The samples were then thawed in a circulating water bath at 37 °C for 30 s, and the content of each straw was poured into a 10 mL conical tube.

### 2.3. Computerized Assisted Semen Analysis

Sperm motility was evaluated by using a computer-assisted sperm analysis (CASA) system (Integrated Sperm Analysis System V1.0; Proiser S.L.; Valencia, Spain). Samples were placed at 37 °C in a water bath for 5 min, and 5 μL of sperm sample was placed onto a slide previously warmed at 37 °C. Samples were then analyzed under a 10× negative phase–contrast objective (Olympus B×41 microscope; Olympus, Tokyo, Japan). Five different fields were analyzed. The following motility descriptors were evaluated: total motility (TM, %), progressive motility (PM, %), curvilinear velocity (VCL, μm/s), straight-line velocity (VSL, μm/s), average path velocity (VAP, μm/s), linearity (LIN, %), straightness (STR, %), oscillation (WOB, %), the amplitude of lateral head displacement (ALH, μm), and frequency of head displacement (BCF, Hz).

CASA settings were: frames per second (25); particle area (> 4 and <75 μm^2^); connectivity (6); minimum number of images to calculate the ALH (10). The cut-off value for motile spermatozoa was VAP ≥ 10 μm/s, and for progressively motile spermatozoa it was STR ≥ 75%

### 2.4. proAKAP4 ELISA Assays

ProAKAP4 analysis was performed in the ONIRIS Laboratory in Nantes (France) using a commercialized sandwich-ELISA-based assay (Horse 4MID^®^ Kit, 4BioDx, Lille, France). The first step was thawing one straw of each ejaculate (See Section 2.2 for more information). Then, 100 µL of post-thawed semen was diluted in the lysis solution, mixed, then loaded onto the 96-well microplate according to the manufacturer’s instructions. A secondary detection antibody was then added to the wells. After washing, the colourimetric substrate was added to each well. This blocked the enzymatic conversion of the colourimetric substrate. The color intensity was measured by a spectrophotometer plate reader, and given optical densities are proportional to proAKAP4 concentrations, based on an internal calibration curve of predetermined proAKAP4 concentration calibrators.

### 2.5. Statistical Analysis

Data were analyzed with the statistical package R (V 4.0.3, R Core Team; Vienna, Austria), and graphs were made with GraphPad Prism software (V 8.4.0, GraphPad Software LLC; San Diego, CA, USA). The first step was to check the normality of the data using the Shapiro-Wilk test, as well as the homoscedasticity of variances using the Levene test. Only when necessary was the function arcsin √x applied to transform the data and, thus, obtain a normal distribution. Then, a hierarchical cluster analysis was performed to classify frozen–thawed horse ejaculates into two groups according to proAKAP4 concentration (low or high). Differences in TM, PM, and motile sperm subpopulations between the two proAKAP4 groups were analyzed using a *t*-test for independent samples. Consequently, A Pearson correlation was applied to obtain the correlation coefficients between proAKAP4 concentration (both groups) with sperm motility parameters (TM, PM, VCL, VSL, VAP, LIN, STR, WOB, ALH, and BCF) and with motile sperm subpopulations of frozen–thawed horse ejaculates. Finally, to analyze proAKAP4 between two quality groups and differences between breeds, a two-way ANOVA was performed.

In all cases, the minimum level of statistical significance was set at *p* ≤ 0.05. Results in the text were expressed as means ± error standard of the mean (SEM).

Motile Sperm Subpopulations

The procedure for calculating the motile sperm subpopulations in frozen–thawed horse ejaculates was the one proposed by Martí et al. [31]. In the first step, a principal component analysis (PCA) was performed with the values of the kinematic parameters (VCL, VSL, VAP, LIN, STR, WOB, ALH, and BCF) of each individual trajectory after thawing. The matrix obtained was rotated using the Varimax method with Kaiser normalization, where the regression scores assigned to each sperm were used to perform a non-hierarchical multivariate cluster analysis using the k-means model based on Euclidean distances calculated from the kinematic parameters. Ultimately, for each frozen–thawed horse ejaculate, the proportion of spermatozoa present in each subpopulation was calculated.

## 3. Results

### 3.1. Classification of Frozen–Thawed Horse Ejaculates according to proAKAP4 Concentration

According to their proAKAP4 concentration, two groups were generated (*p* < 0.001) as having low (21.20 ± 1.94 ng/10M spz; range = 5.06–35.61 ng/10M spz; n = 23) or high (54.79 ± 2.35 ng/10M spz; range = 39.92–82.23 ng/10M spz; n = 25) proAKAP4 concentration (Figure 1).

### 3.2. Relationship between proAKAP4 Groups and Its Distribution in Different Breeds

Taking account different breeds in our study, there are not significant different between breeds (*p* > 0.05). The Arabian pure breed (n = 8 ejaculates) showed a median of 53.31 ± 4.00 ng/10M spz in the high quality group (n = 6) and a median of 19.75 ± 0.96 ng/10M spz in the low quality group (n = 2). The pure Spanish breed (n = 33 ejaculates) showed a median of 57.64 ± 3.28 ng/10M spz in the high quality group (n = 15) and, in the low quality group (n = 18), a median of 19.75 ± 19.51 ng/10M spz. The Warmblood breed (n = 7 ejaculates) showed a median of 46.31 ± 3.41 ng/10M spz in the high quality group (n = 4) and, in the low quality group (n = 3), a median of 32.28 ± 2.29 ng/10M spz (Figure 2).

### 3.3. Relationship between proAKAP4 Concentration and Sperm Motility and Motile Sperm Subpopulations in Frozen–Thawed Horse Ejaculates

The percentage of total motile sperm was significantly lower (*p* ≤ 0.05) in ejaculates having low (62.91 ± 4.60%) proAKAP4 concentration compared to those having high (73.06 ± 2.65%) concentration (Figure 3a). On the contrary, the percentage of progressive motile sperm did not differ between frozen–thawed ejaculates with low or high proAKAP4 concentration (Figure 3b).

Four sperm motile subpopulations were identified in frozen–thawed horse ejaculates (Table 1). Subpopulation 1 (SP1) was the fastest (higher values of VCL, VSL, and VAP), with intermediate values of LIN, STR, and WOB, and with ALH also intermediate and BCF higher. Subpopulation 2 (SP2) presented intermediate speed values, but was the most progressive, with higher values of LIN, STR, and WOB, and with intermediate ALH and BCF. Subpopulation 3 (SP3) was the slowest, with intermediate values of LIN, STR, and WOB, and with the lowest ALH and BCF values. Finally, subpopulation 4 (SP4) also presented intermediate speed values, but it showed low values of LIN, STR and WOB, although it had the highest ALH and intermediate BCF.

The proportion of motile sperm of SP3 identified in frozen–thawed horse ejaculates (Figure 4) was significantly higher (*p* < 0.05) in ejaculates having low (47.40 ± 4.46%) proAKAP4 concentration compared to those having high (37.16 ± 2.83%) concentration. On the contrary, SP1, SP2, and SP4 did not show differences between ejaculates as having low or high proAKAP4 concentrations.

### 3.4. Correlation between proAKAP4 Concentration with Sperm Motility and Motile Sperm Subpopulations of Frozen–Thawed Horse Ejaculates

The correlation coefficients between proAKAP4 concentration with sperm motility and motile sperm subpopulations of frozen–thawed horse ejaculates are shown in Table 2. ProAKAP4 concentration shows a positive correlation with TM (r = 0.31; *p* < 0.05; Table 2), with PM (r = 0.31; *p* < 0.05), as well as with kinematic parameters of speed: VCL (r = 0.29; *p* < 0.05), VSL (r = 0.34; *p* < 0.05), and VAP (r = 0.33; *p* < 0.05). On the contrary, proAKAP4 concentration was negatively correlated with SP3 (r = –0.37; *p* < 0.05). While, with SP1, SP2, and SP4, there was a positive correlation, although not significant (Table 2).

## 4. Discussion

Classification of frozen/thawed stallion ejaculates depending on proAKAP4 concentration has been established. Stallion ejaculates with high proAKAP4 concentrations are inside the rank of 39.92–82.23 ng/10M spz. Those having low proAKAP4 concentrations are between 5.06–35.61 ng/10M spz. This can establish a mean of 37.77 ng/10M spz. In the same way, when we compare the three breeds of this study (Arabian, Spanish, and Warmblood), quality groups are consistent. Our results agree with 4BioDx commercial information [32], considering the rank 0–15 ng/10M spz bad semen, 15–40 ng/10M spz correct semen, and more than 40 ng/10M spz very good semen. The mean concentration obtained in the present study is then in agreement with the established thresholds of proAKAP4 described in other mammal species. Herein, sperm having more than 37.77 ng/10M spz could be considered as very good semen, whereas those having lower concentration, or good, or with concentration below 15 ng/10M spz, are of low quality.

As a precursor of the AKAP4 protein, proAKAP4 is positively correlated with total and progressive motility; the same results were found [24,28] in stallions, but also in bulls [33,34], men [35], mice [26], dromedaries [19], and rams [16]. A deeper analysis of the motility descriptors, proAKAP4 shows a positive correlation with velocity parameters (VCL, VSL, and VAP), but not with pathway descriptors (LIN, STR, WOB, ALH, and BCF). By the analysis of these results, we can conclude that proAKAP4 describes the capacity of spermatozoa to move forward through the velocity descriptors, but it does not describe how this movement occurs. Further studies are needed to identify how proAKAP4 affects the trajectory of the spermatozoa.

While considering the relationship between subpopulations and proAKAP4 concentration, subpopulation 3 (SP3) is higher in ejaculates of the lowest proAKAP4 concentration group. This result is supported by a negative correlation between proAKAP4 and SP3 found when taking all the ejaculates together. This means that, the slower the spermatozoa are, the lower the proAKAP4 concentration is. In consequence, one ejaculate with a high percentage of slow spermatozoa will show reduced proAKAP4 levels. Other subpopulations are positively correlated with proAKAP4 concentration, even if it is not statistically significant. This confirms that proAKAP4 is related to the motion of the spermatozoa [9,10,24]. The first time when spermatic subpopulations in stallions were described [3], four subpopulations were determined, as in our study. A low proAKAP4 concentration could be an indicator of early semen deterioration because of a high percentage of SP3. In an ejaculate, this means a descent of quality parameters such as viability or total motility [3]. This result is complementary and agrees with Griffin et al. [28], who found a positive correlation between proAKAP4 and the percentage of rapid spermatozoa in the ejaculate.

Boersma et al. [36] demonstrated that proAKAP4 is not affected by cryopreservation in epidydimary semen in mice, which is relvevant because proAKAP4 in frozen/thawed semen can be a good predictor of its quality before ejaculation. Nevertheless, it is important to keep in mind that epidydimary sperm has not been under the influence of the seminal plasma. Further studies are needed to elucidate how cryopreservation affects proAKAP4 concentration in stallion-ejaculated sperm.

The importance of spermatic motility on fertility rates has been described over the years [37,38,39,40,41,42,43], but some stallions are not fertile even if this is not reflected in motility [37,44]. The consequence of motility on fertility is not clear [45,46]. ProAKAP4 describes the motion of the spermatozoa [9,10,24], but not how they move. This may evoke a bias relating the proAKAP4 and fertility. AKAP4 has been elucidated as having a role in the phosphorylation before capacitation [28] due to protein translocation at the plasma membrane reorganization during the capacitation process, including hypermotility [47]. Ramal-Sanchez et al. [48] have shown that spermatozoa from bovine oviductal epithelial cells after progesterone influence showed increased levels of AKAP-4, and, at this moment, spermatozoa show a hypermotility pattern [48,49]. AKAP4 can be considered as a possible biomarker of fertility in bulls [48]. Further studies need to be performed with proAKAP4 to relate its action mechanism with motility patterns and the outcome of fertility.

## 5. Conclusions

The concentration of proAKAP4 after freezing/thawing in stallion sperm correlates positively with total and progressive motility, as well as with velocity kinematic parameters. In addition to this, a negative relationship was observed between ejaculates with proAKAP4 levels and SP3, which is the slowest motile sperm subpopulation. Concentrations of proAKAP4 higher than 37.77 ng/10M spz are correlated with qualified frozen/thawed semen.

## Figures and Tables

**Figure 1 animals-12-03417-f001:**
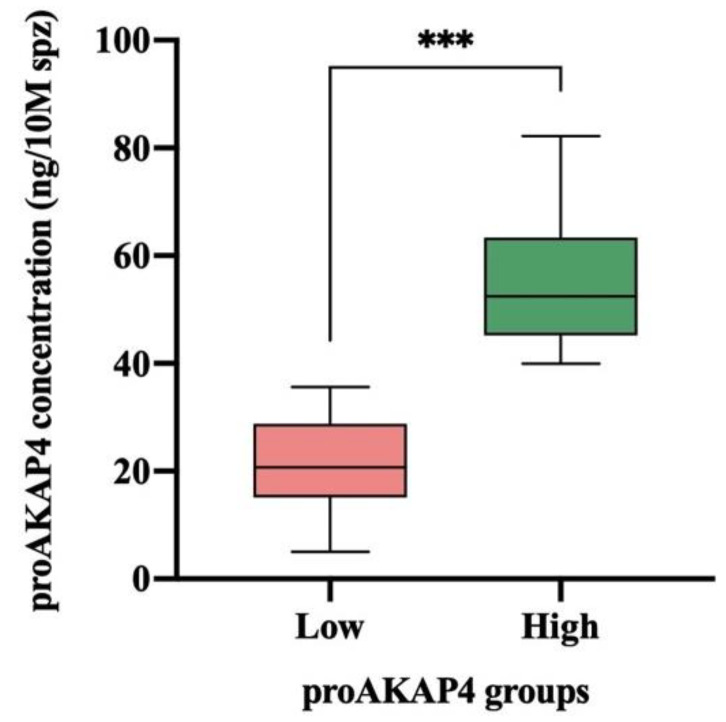
Box–whisker plot showing the proAKAP4 concentration (ng/10M spz) of frozen–thawed horse ejaculates as having low (n = 23; red) or high (n = 25; green) proAKAP4 concentrations. The line indicates the median, the boxes enclose the 25th and 75th percentiles, and the whiskers extend to the 5% and 95% percentiles (*** *p* ≤ 0.001).

**Figure 2 animals-12-03417-f002:**
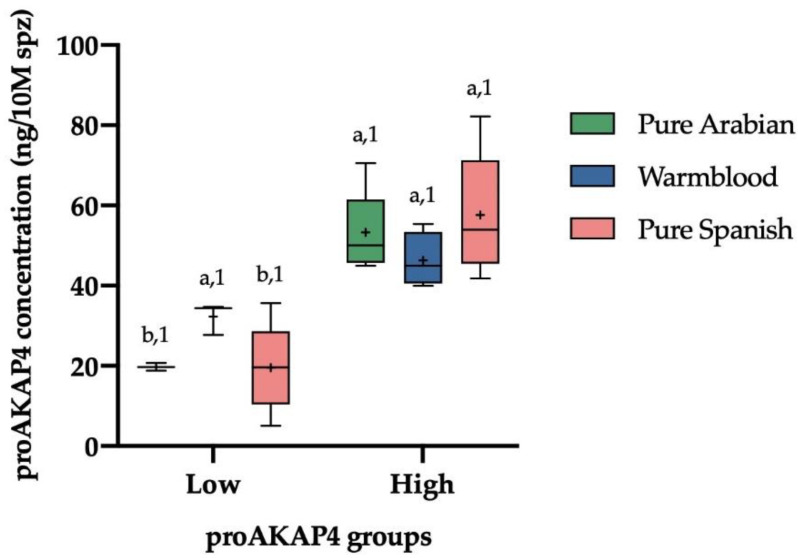
Box–whisker plot showing the differences in proAKAP4 concentration (ng/10M spz) of frozen–thawed horse ejaculates from different breeds (Pure Arabian breed = red; Warmblood breed = blue; pure Spanish breed = green) as having low or high proAKAP4 concentration (ng/10M spz). The line indicates the median, the cross indicates the mean, the boxes enclose the 25th and 75th percentiles, and the whiskers extend to the 5th and 95th percentiles. (a, b) Different letters indicate significant difference (*p* ≤ 0.05) between the proAKAP4 groups within each breed. The same number indicates no significant difference (*p* > 0.05) between breeds within each proAKAP4 group.

**Figure 3 animals-12-03417-f003:**
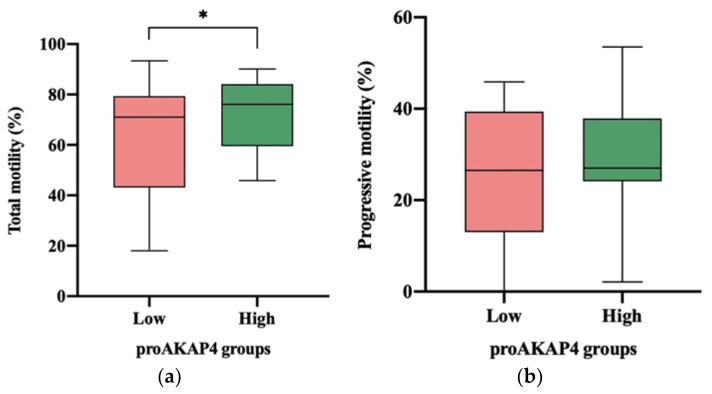
Box–whisker plot showing the differences in total (**a**) and progressive (**b**) motility of frozen–thawed horse ejaculates as having low (n = 23; red) or high (n = 25; green) proAKAP4 concentration (ng/10M spz). The line indicates the median, the boxes enclose the 25th and 75th percentiles, and the whiskers extend to the 5th and 95th percentiles. * *p* ≤ 0.05.

**Figure 4 animals-12-03417-f004:**
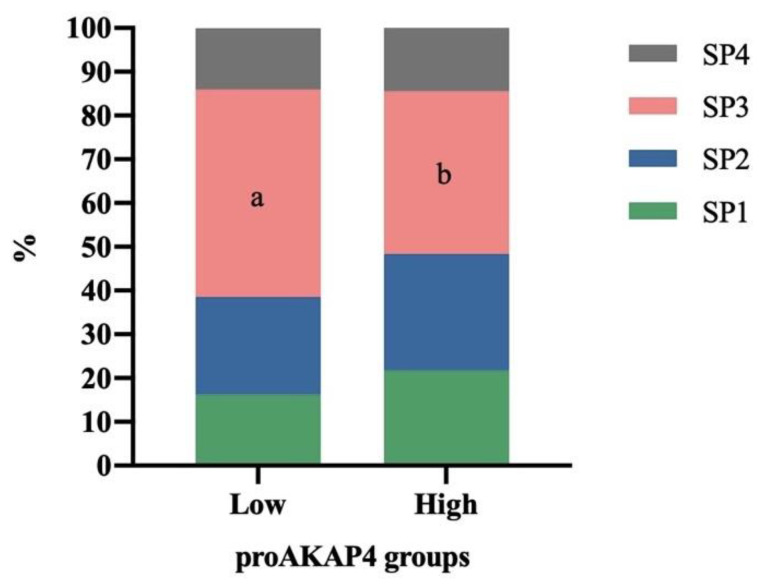
Distribution of the four motile sperm subpopulations (SP) identified in frozen–thawed horse ejaculates as having low (n = 23) or high (n = 25) proAKAP4 concentration (ng/10M spz). a, b Different letters indicate significant differences (*p* ≤ 0.05) between proAKAP4 groups.

**Table 1 animals-12-03417-t001:** Structure (mean ± SEM; range) of the four motile sperm subpopulations (SP) identified in frozen–thawed horse ejaculates (n = 48).

Parameter	SP1		SP2		SP3		SP4	
	Mean ± SEM	Range	Mean ± SEM	Range	Mean ± SEM	Range	Mean ± SEM	Range
VCL (µm/s)	113.00 ± 0.40	83.86–236.75	66.16 ± 0.27	21.45–96.97	29.92 ± 0.18	10.01–70.65	89.59 ± 0.52	37.37–211.58
VSL (µm/s)	64.39 ± 0.28	29.48–121.82	44.21 ± 0.20	20.41–77.71	12.03 ± 0.09	0.00–27.82	21.61 ± 0.22	0.00–41.60
VAP (µm/s)	77.93 ± 0.27	45.35–141.79	49.72 ± 0.21	21.07–80.18	17.56 ± 0.11	5.14–42.53	48.93 ± 0.33	11.10–108.92
LIN (%)	58.60 ± 0.31	14.94–96.66	68.18 ± 0.24	38.82–98.31	42.74 ± 0.25	0.00–98.38	24.20 ± 0.23	0.00–46.06
STR (%)	82.95 ± 0.25	27.79–99.47	89.05 ± 0.14	49.47–99.65	69.07 ± 0.28	0.00–99.26	44.42 ± 0.40	0.00–92.91
WOB (%)	70.13 ± 0.26	28.38–100.00	76.35 ± 0.22	43.26–100.00	60.89 ± 0.21	16.44–100.00	55.06 ± 0.26	16.70–97.47
ALH (µm)	3.83 ± 0.03	0.78–10.54	2.31 ± 0.01	0.49–5.97	1.47 ± 0.01	0.23–4.79	3.97 ± 0.03	1.21–11.70
BCF (Hz)	12.26 ± 0.08	0.00–22.00	10.67 ± 0.06	0.00–21.00	6.99 ± 0.04	0.00–19.00	8.34 ± 0.07	0.00–21.00
**n (%)**	2517 (19.00)		3293 (24.86)		5097 (38.47)		2341 (17.67)	

VCL (µm/s): curvilinear velocity; VSL (µm/s): straight-line velocity; VAP (µm/s): average path velocity; LIN (%): linearity coefficient; STR (%): straightness coefficient; WOB (%): wobble coefficient; ALH (µm): amplitude of lateral head displacement; BCF (Hz): beat-cross frequency; SEM: standard error of the mean; n: number of spermatozoa; %: percentage of subpopulation.

**Table 2 animals-12-03417-t002:** Correlation coefficients between proAKAP4 concentration with sperm motility and motile sperm subpopulations of frozen–thawed horse ejaculates (n = 48).

Parameter	proAKAP4 Concentration (ng/10 M spz)
MT (%)	0.31 (*p* = 0.03)
MP (%)	0.31 (*p* = 0.03)
VCL (µm/s)	0.29 (*p* = 0.04)
VSL (µm/s)	0.34 (*p* = 0.02)
VAP (µm/s)	0.33 (*p* = 0.02)
LIN (%)	0.15
STR (%)	0.19
WOB (%)	0.01
ALH (µm)	0.20
BCF (Hz)	0.27
SP1 (%)	0.26
SP2 (%)	0.25
SP3 (%)	–0.37 (*p* = 0.01)
SP4 (%)	0.15

TM (%): total motility; PM (%): progressive motility; VCL (µm/s): curvilinear velocity; VSL (µm/s): straight-line velocity; VAP (µm/s): average path velocity; LIN (%): linearity coefficient; STR (%): straightness coefficient; WOB (%): wobble coefficient; ALH (µm): amplitude of lateral head displacement; BCF (Hz): beat-cross frequency; SP (%): subpopulation.

## Data Availability

“MDPI Research Data Policies” at https://www.mdpi.com/ethics.

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
