# Peer review of "ProAKAP4 Concentration Is Related to Sperm Motility and Motile Sperm Subpopulations in Frozen–Thawed Horse Semen"

_animals, 2022, doi:10.3390/ani12233417_

Round 1
Reviewer 1 Report
The manuscript from Dordas-Perpinyà and coll., entitled “ProAKAP4 concentration is related with sperm motility and motile sperm subpopulations in frozen-thawed horse semen” and submitted to Animals investigated the potential relationship between the precursor proAKAP4 and horse sperm motility. Manuscript would be accepted after minor revision, and some comments and suggestions are listed below. The manuscript needs an English revision (in terms of grammar and styling).
Comments, questions and suggestions:
- Considering that multiple ejaculates were obtained from the same animal, and that the number of animals is 13, do the Authors compare the proAKAP4 concentration among the animals, instead of comparing among ejaculates? Is there any correlation? In this R opinion, some lines could be added regarding this issue.
-Abstract should be revised. Please, refer to the guidelines for the manuscript writing and include the background prior to introducing the aim of the work.
(Abstract: The abstract should be a total of about 200 words maximum. The abstract should be a single paragraph and should follow the style of structured abstracts, but without headings: 1) Background: Place the question addressed in a broad context and highlight the purpose of the study; 2) Methods: Describe briefly the main methods or treatments applied. Include any relevant preregistration numbers, and species and strains of any animals used. 3) Results: Summarize the article's main findings; and 4) Conclusion: Indicate the main conclusions or interpretations. The abstract should be an objective representation of the article: it must not contain results which are not presented and substantiated in the main text and should not exaggerate the main conclusions).
-332 please rewrite this sentence
-Do the Authors have any hypothesis regarding the relationship between proAKAP4 and AKAP4, and how the excision of the aas could interfere or cause modifications in the sperm motility/hypermotility? This R would appreciate an hypothesis about the results obtained an a discussion about it.
-Just as a curiosity, in a vitro study co-authored by this Reviewer, AKAP-4 was found to be more abundant in spermatozoa released from oviductal epithelial cells with or without the action of progesterone (bovine model, results obtained by proteomic analysis). This Reviewer finds this new manuscript interesting due to obvious reasons.
Among the differential proteins, those with the highest fold-changes in the BOEC-P4 vs. CTRL comparison were flagellar proteins: the AKinase anchoring protein 4 (AKAP4; BOEC-P4:CTRL ratio of 4.2) and dynein heavy chain 7 (DNAH7; ratio of 3.6), both more abundant in the BOEC-P4 and in the BOEC groups compared with controls (BOEC:CTRL ratios of 2.7 and 2.8, respectively). AKAP4 and DNAH7 are major components of the sperm fibrous sheath (i.e., the extent of the principal piece region of the sperm flagellum) that plays important roles in sperm motility (Baccetti et al., 2005; Moretti et al., 2007). Furthermore, AKAP-4 is involved in the cAMP/PKA and PKC/ERK1-2 signaling pathways leading to tyrosine phosphorylation, actin polymerization and acquisition of sperm motility (Rahamim Ben-Navi et al., 2016). AKAP-4 is the mature form obtained from pro-AKAP-4 after the removal of 188 amino acids from the N-terminal domain, which allows AKAP-4 to bind AKAP-3 by the C-terminal domain. In the present study, one of the AKAP-4 peptide (of 4960.6 Da) that was found to be more abundant in the BOEC-P4 group compared with controls corresponds to this C-terminal region of AKAP-4. These results are concordant with those found by others authors (Romero-Aguirregomezcorta et al., 2019), showing an increase in sperm hypermotility at the time of the sperm release from BOECs by the action of P4. (Extracted from Ramal-Sanchez et al 2020, Mol. Cell. Endo.)
Author Response
REVIEWER 1
Thank you very much for your comments they has helped us to improve our manuscript.
Comment 1: Considering that multiple ejaculates were obtained from the same animal, and that the number of animals is 13, do the Authors compare the proAKAP4 concentration among the animals, instead of comparing among ejaculates? Is there any correlation? In this R opinion, some lines could be added regarding this issue.
Answer: We consider the ejaculate as the experimental unit, regardless of whether it comes from the same animal, so the analyzes performed are not of the animals but of the ejaculates because each ejaculate is classified as high or lowconcentration of proAKAP4, not the animal.
Comment 2: Abstract should be revised. Please, refer to the guidelines for the manuscript writing and include the background prior to introducing the aim of the work.
(Abstract: The abstract should be a total of about 200 words maximum. The abstract should be a single paragraph and should follow the style of structured abstracts, but without headings: 1) Background: Place the question addressed in a broad context and highlight the purpose of the study; 2) Methods: Describe briefly the main methods or treatments applied. Include any relevant preregistration numbers, and species and strains of any animals used. 3) Results: Summarize the article's main findings; and 4) Conclusion: Indicate the main conclusions or interpretations. The abstract should be an objective representation of the article: it must not contain results which are not presented and substantiated in the main text and should not exaggerate the main conclusions).
Answer: Thanks for your suggestion. The abstract was revised and changed accordingly the guide of authors and the background was included.
Comment 3: -332 please rewrite this sentence
Answer: The sentence was deleted. We think the text is better without this sentence.
Comment 4: Do the Authors have any hypothesis regarding the relationship between proAKAP4 and AKAP4, and how the excision of the aas could interfere or cause modifications in the sperm motility/hypermotility? This R would appreciate an hypothesis about the results obtained an a discussion about it.
Answer: We have the hypothesis that AKAP4, and proAKAP4 as its reservoir, plays an important role in the hypermotility as some references in the paper confirm it. On the contrary, we have not thought about the aas modifications during motility/hypermotility because after the phosphorylation of proAKAP4 exists a translocation of the AKAP4 in the axoneme in human during the capacitation process, including hypermotility (Ben-Navi et al. 2016) and we would like to keep working on this in animal spermatozoa. A sentence about this hypothesis has been added to the discussion and we think than it makes the paper and conclusion much clear.
Comment 5: Just as a curiosity, in a vitro study co-authored by this Reviewer, AKAP-4 was found to be more abundant in spermatozoa released from oviductal epithelial cells with or without the action of progesterone (bovine model, results obtained by proteomic analysis). This Reviewer finds this new manuscript interesting due to obvious reasons.
Among the differential proteins, those with the highest fold-changes in the BOEC-P4 vs. CTRL comparison were flagellar proteins: the AKinase anchoring protein 4 (AKAP4; BOEC-P4:CTRL ratio of 4.2) and dynein heavy chain 7 (DNAH7; ratio of 3.6), both more abundant in the BOEC-P4 and in the BOEC groups compared with controls (BOEC:CTRL ratios of 2.7 and 2.8, respectively). AKAP4 and DNAH7 are major components of the sperm fibrous sheath (i.e., the extent of the principal piece region of the sperm flagellum) that plays important roles in sperm motility (Baccetti et al., 2005; Moretti et al., 2007). Furthermore, AKAP-4 is involved in the cAMP/PKA and PKC/ERK1-2 signaling pathways leading to tyrosine phosphorylation, actin polymerization and acquisition of sperm motility (Rahamim Ben-Navi et al., 2016). AKAP-4 is the mature form obtained from pro-AKAP-4 after the removal of 188 amino acids from the N-terminal domain, which allows AKAP-4 to bind AKAP-3 by the C-terminal domain. In the present study, one of the AKAP-4 peptide (of 4960.6 Da) that was found to be more abundant in the BOEC-P4 group compared with controls corresponds to this C-terminal region of AKAP-4. These results are concordant with those found by others authors (Romero-Aguirregomezcorta et al., 2019), showing an increase in sperm hypermotility at the time of the sperm release from BOECs by the action of P4. (Extracted from Ramal-Sanchez et al 2020, Mol. Cell. Endo.)
Answer: Thank you very much for this comment, yes we have the hypothesis that AKAP4 has an important role in hypermotility then your conclusion is so important to keep working on it. Then, we have incorporated these 2 references in the discussion.
Reviewer 2 Report
In general, the paper explores an interesting question regarding proAKAP4 and sperm quality. The manuscript would benefit from a thorough edit for English. In addition, please consider the following suggestions to improve the quality of the manuscript:
Line 117: Please confirm that 60% viability is the minimum threshold for standard acceptable viability. In addition, it would be good to clarify the inclusion for “good” vs “poor” semen here.
Lines 115-117: Please clarify in the text how viability was measured.
Lines 135-150: Was viability post-thaw evaluated?
Statistics: The grouping of “high” and “low” proAKAP4 seems artificial, and it may be more appropriate to use a linear regression model to investigate the relationship between proAKAP4 and motility.
Line 176 and elsewhere: Please confirm the threshold for significance. As written, p=0.05 and below is considered significantly different. Typically, p<0.05 is used. Please confirm if it was your intention to have “less than or equal to” or “less than”.
Author Response
REVIEWER 2
General Comment: In general, the paper explores an interesting question regarding proAKAP4 and sperm quality. The manuscript would benefit from a thorough edit for English. In addition, please consider the following suggestions to improve the quality of the manuscript:
Answer: Dear reviewer, thanks for your kind review and comment that help us to improve our article.
Comment 1: Line 117: Please confirm that 60% viability is the minimum threshold for standard acceptable viability. In addition, it would be good to clarify the inclusion for “good” vs “poor” semen here.
Answer: This paragraph was deleted because is confused. As explained later samples were classified by a hierarchical cluster analysis to identify “high” and “low” proAKAP4 and then the groups were compared considering the sperm motility descriptors and subpopulations.
Comment 2: Lines 115-117: Please clarify in the text how viability was measured.
Answer: Viability was measured by a Eosin-Nigrosin smears as is explained in the reviewed manuscript.
Comment 3: Lines 135-150: Was viability post-thaw evaluated?
Answer: Yes, post-thaw sperm viability was evaluated. However, only significant correlation were observed between sperm motility and Proakap4, as explained in the text.
Comment 4: Statistics: The grouping of “high” and “low” proAKAP4 seems artificial, and it may be more appropriate to use a linear regression model to investigate the relationship between proAKAP4 and motility.
Answer: The ejaculates classification is not artificial. We did a hierarchical cluster analysis to identify “high” and “low” proAKAP4 groups taking into account the individual proAKAP4 concentration. And then, as explained in material and methods, CASA motility descriptors and sperm subpopulations were compared between groups. The objective was to stablish a cut-of.
On the other hand, we did a linear regression. Obtained results confirmed previous results obtained by the hierarchical cluster. We can include this information as supplementary data.
Comment 5: Line 176 and elsewhere: Please confirm the threshold for significance. As written, p=0.05 and below is considered significantly different. Typically, p<0.05 is used. Please confirm if it was your intention to have “less than or equal to” or “less than”.
Answer: Sorry it’s a mistake. p<0.05 in all cases.
Round 2
Reviewer 2 Report
The manuscript is improved and more clear. Please consider the following suggestions prior to pubilcation:
Please edit this article for English, as there are grammatical errors throughout. As an example:
Line 47: The text should read “Spermatozoa are motile cells” rather than “Spermatozoa is a motile cell”
Line 65: The text should read “the principal piece drives” not “the principal piece drive”
Line 70: The text should read “tuned” rather than “tune”
Line 83: The text should read “reflects” rather than “reflect”
There are others as well, and a thorough edit would help a lot.
Author Response
Dear reviewer thanks for your kind review.
All of your comments were corrected and a native has done a general review of the document.